# Subsequent memory effect in the inferior frontal gyrus revealed by fNIRS

**Petra Bíró, Silvy H. P. Collin**🅾*

Department of Computational Cognitive Science, Tilburg School of Humanities and Digital Sciences, Tilburg University, Tilburg, The Netherlands

* s.h.p.collin@tilburguniversity.edu

## Abstract

A central finding in memory research is the subsequent memory effect, which describes consistent neural differences during encoding, where later remembered events elicit higher brain activity than those that are later forgotten, which has been reliably replicated with both EEG and fMRI for the past decades. Replicating the subsequent memory effect using fNIRS could enable research opportunities that are difficult to pursue with other methods, including studies with children, patient populations, or experiments in highly naturalistic settings. Therefore, our study investigated whether the prefrontal cortex is differentially involved in subsequently remembered versus subsequently forgotten stimuli using fNIRS. Our results showed that in particular channels mapping onto the inferior frontal gyrus showed more activation during encoding of subsequently remembered events compared to subsequently forgotten events. These results demonstrate that fNIRS can reliably capture the subsequent memory effect, providing new opportunities to study memory mechanisms across diverse populations and real-world contexts.

## 1. Introduction

Understanding why some experiences are remembered while others are forgotten is a central question in cognitive neuroscience [1,2], as well as studying what characteristics define successful memory formation. Relating what happens in the brain during an experience to later behavioral evidence – which tells us that memory was created – can give us insight into this question. Memory formation has been shown to have at least two components: the first mediates the transformation of sensory input into internal representations, and the second turns these representations into a long lasting trace [3]. There is evidence that this process relies on the interaction between the prefrontal cortex (PFC) and the medial temporal lobe (MTL) [4]. The subsequent memory effect is one of the core findings in memory research that has been utilized to study how memories are formed. It refers to the reliable neural differences

**Data availability statement:** The data is now publicly available and the statement included reads as follows: Code and data are available at the Tilburg University Dataverse repository called Code and data of Biro and Collin Subsequent memory effect in the inferior frontal gyrus revealed by fNIRS: https://doi.org/10.34894/A0OQMU.

**Funding:** The author(s) received no specific funding for this work.

**Competing interests:** The authors have declared that no competing interests exist.

observed at encoding between events that are later remembered and those that are later forgotten; where neural activity is higher for later remembered events [5–8].

In a subsequent memory paradigm, encoding trials are categorized as remembered or forgotten on the basis of later memory performance. Brain activity measured during encoding is then split into later remembered and later forgotten categories, and the goal is to measure whether brain activity is higher in the remembered condition. The first fMRI studies demonstrating a subsequent memory effect [9,10] found that activity was higher for subsequently remembered trials, compared to forgotten trials in the left inferior frontal gyrus (IFG), the temporal cortex, and parahippocampal regions. The effect has been robustly replicated across methodologies; EEG [11–16], fMRI [17–22], and combined EEG-fMRI [23,24]. A meta-analysis [8] found that subsequent memory effects are most prominent in the inferior frontal gyrus (IFG). Several other fMRI studies also reported on activation in the left IFG during encoding [25,26]. Additionally, subsequent memory effects have been found in other subregions within the PFC, like dorsolateral PFC [22] and ventromedial PFC [20,27]. Clinically, the subsequent memory effect has also been investigated in individuals from several types of patient groups that are known to (also) have memory-related symptoms, like Alzheimer's disease patients [28], schizophrenia patients [29] and epilepsy patients [26].

A more data-driven approach to studying memory formation involves measuring neural synchrony across individuals experiencing the same event, such as listening to the same story [30]. Although memory formation inherently involves individual differences, it remains a compelling question whether shared neural patterns across participants during encoding predict subsequent memory success. Inter-subject correlation (ISC) analyses have been used to address this, examining whether remembered events elicit greater neural synchrony than forgotten ones. fMRI studies [31,32] have identified subsequent memory effects (SME) using ISC in the (v)mPFC. Notably, [32] found a reversed effect in the IFG, with no neural synchrony present across participants.

Building on this foundation, our project investigates the feasibility of using a relatively new neuroimaging method –functional Near-Infrared Spectroscopy (fNIRS)– to reliably detect such a subsequent memory effect in the human brain. fNIRS is a tool that measures hemodynamic response and primarily differs from other brain activity measurement techniques in its portability, making it particularly attractive for cognitive neuroscience studies aiming to explore real-world functioning. fNIRS has been used to measure memory effects, such as episodic memory encoding and recall [33], the effect of age on memory recall in the PFC [34], how music impacts encoding and recall [35], influence of working memory related processing [36–38]. [39] measured a construct closely related to SME, using the directed forgetting paradigm, where subjects were instructed to remember or forget a certain stimulus. Behavioral responses were correlated with fNIRS-measured brain activity during encoding trials. No significant differences were observed in the 0–5s window; however, the 5–9s window revealed significantly greater activation for intentional remembering than intentional forgetting in the right angular gyrus. Although this paradigm demonstrates that fNIRS

can differentiate between these conditions, to our knowledge, no substantial research has examined the subsequent memory effect using fNIRS.

In summary, both fMRI and EEG have been used to study the subsequent memory effect or similar paradigms, and while fNIRS has been used in memory research, there is a lack of substantial research directly investigating the subsequent memory effect using fNIRS. Our study addresses this gap by investigating whether fNIRS measurements during memory encoding can reliably detect the difference between later remembered and forgotten events. We focus specifically on the PFC, since fNIRS is constrained to cortical regions, and memory formation – especially the initial component of it – involves the PFC. Various individual studies using in particular fMRI have discovered subsequent memory effects in various subregions within the PFC (e.g., [22] in dorsolateral PFC, and [20] in medial PFC). While SME effects have also been reported in non-prefrontal regions such as the fusiform cortex and posterior parietal cortex [8], these fall outside the spatial scope of our fNIRS setup. We therefore expected the strongest subsequent memory effect in the IFG, given that a large meta-analysis – across as many as 74 neuroimaging studies [8] – most consistently implicates this part of the PFC in a subsequent memory effect. IFG is thought to support semantic elaboration during encoding [40], a process by which new information is linked to existing knowledge and which is considered a key mechanism underlying successful memory formation [41]. Accordingly, our ISC analysis will also focus on the IFG, investigating whether fNIRS can capture ISC-based SME effects in the PFC, and allowing us to approach the subsequent memory effect from both an individual and a cross-participant perspective. Translating the subsequent memory effect — robustly demonstrated with other neuroimaging methods – to fNIRS will open avenues for research not easily performed with other methods like fMRI [42], such as studies in children, patient groups, or highly naturalistic paradigms.

## 2. Materials and methods

Research Ethics and Data Management Committee of the Tilburg School of Humanities and Data Sciences of Tilburg University has approved the study (approval number REDC2024.08a). We obtained written informed consent from all participants. This study is a re-analysis of the data from [43] which focused on how the PFC is involved in representing schema-violating information. The present study addresses a distinct research question focused on the subsequent memory effect. For this purpose, we re-used the data from day 1 of this fNIRS experiment to investigate whether the PFC was involved in subsequent memory performance.

### 2.1. Participants

The dataset has 38 participants between 18 and 33 years old. Participants had no diagnosed neurological or psychiatric disorders. Data collection was performed at the DAF Technology lab between 14 November 2024 and 15 February 2025. Participants gave written informed consent before starting the experiment, and received study credits for their participation. Two participants were excluded due to technical issues during data acquisition. The experiment was approved by the Research Ethics and Data Management Committee of TSHD, Tilburg University.

### 2.2. Summary of task, stimuli and data preprocessing

Here, a summary of the task that participants performed on day 1 of this 2-day experiment, the stimuli used, and the data preprocessing steps taken. More details can be found in [43].

In this experiment, participants were seated in front of a computer monitor wearing a g.tec fNIRS headcap [44] and encoded 32 individual stimuli (that on day 1 lasted 9 seconds per stimulus, with an 1 second intertrial interval) (Fig 1).

These stimuli were fictional activity descriptions, each related to either a holiday or educational activity. An example of such an activity description is: *Here, people cultivate and observe plants specifically adapted to thrive in lunar environments, considering low gravity and limited resources.* Besides reading the activity description, participants also viewed a corresponding map with the activity's location circled. Descriptions were shown below the map with a title above, and all

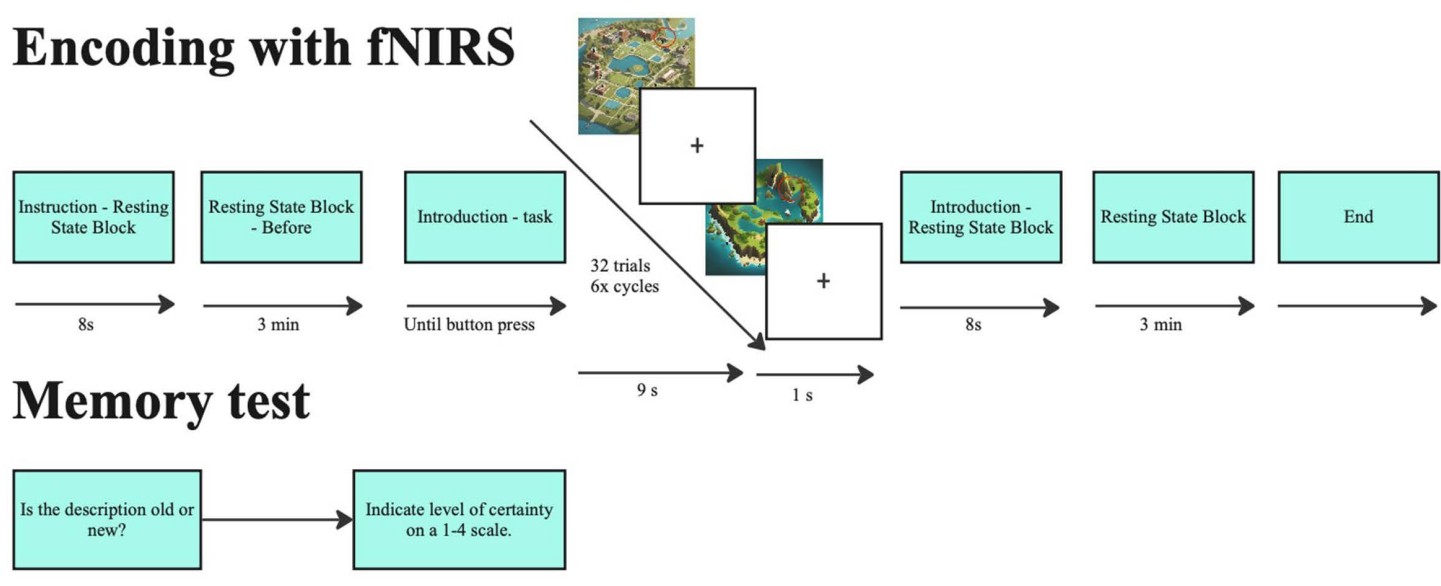

**Fig 1. Study procedure.**

were created using ChatGPT and manually refined. The participants were instructed to do the following: 'In the following task you will read descriptions of various activities. Each activity will be shown along with one out of two maps. The map indicates at which location that particular activity happened. Please read the text carefully and try to remember the text and the locations of the activities.'

Participants' memory for the stimuli was tested in a subsequent recognition memory test (self-paced) directly following encoding that included the actual stimuli, 32 lure stimuli and 32 completely new stimuli. In this memory test, participants were asked whether they recognized the description (i.e., old or new), whether they remember which map the stimulus was from (holiday or education) and whether they remember which location the stimulus was from (out of 9 possible location on a map). As memory performance measure, recognition of the activity descriptions was used (i.e., the old/new question).

fNIRS signals were recorded wirelessly using the g.Nautilus system with eight frontal cortex optodes at 250 Hz, using 760 nm and 850 nm LEDs, synchronized via Bluetooth with Simulink/Matlab (Fig 2). fNIRS data were preprocessed using MNE [45], including bandpass filtering for 0.01–0.1 Hz [46], epoching (0–8 s), downsampling to 10 Hz, and baseline correction using the rest block preceding the encoding task itself; E-prime trial sequences were exported, and added as epoch metadata for analysis in R.

Our preprocessing pipeline for both fNIRS and behavioral data was identical to [43]. More details concerning the data preprocessing can therefore be found in the methods section of [43].

## 2.3. Statistical analysis

To address our research question, binary logistic regression was employed [47] to examine whether brain activity measured through fNIRS could predict subsequent memory performance. The dependent variable in the analysis, *remembered_forgotten*, indicates whether a participant recalled a specific item or not. The independent variable, *value*, represents the continuous HbO (oxy-hemoglobin) signal recorded during encoding on day 1. The fNIRS HbO signal has been shown to correlate with the fMRI BOLD response, as both reflect neurovascular coupling [48]. A binary logistic regression model was run on all channels. Due to the delayed peak of the BOLD related fNIRS data, and to take into

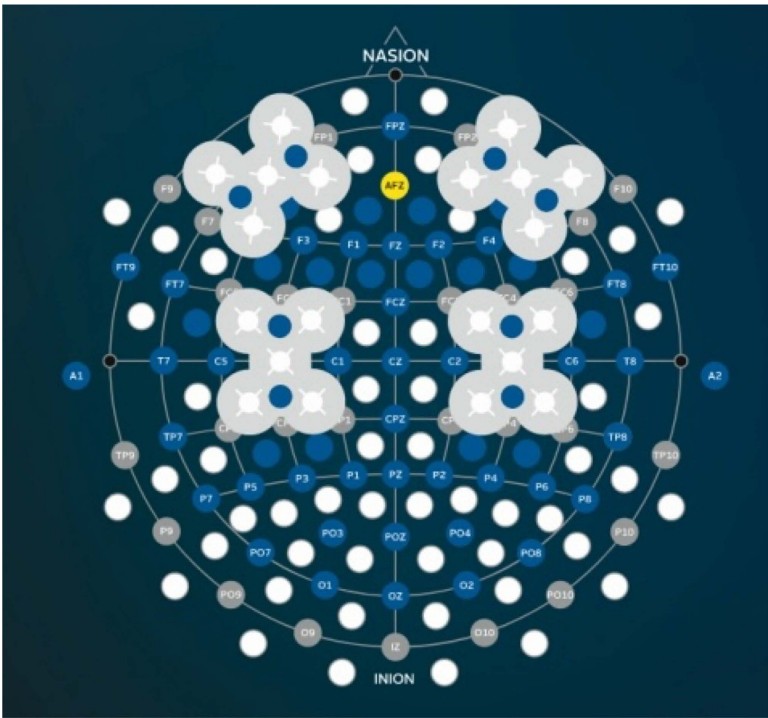

**Fig 2. fNIRS transmitter placement. g.tec medical engineering GmbH, 2025.** Used with permission of the rights holder. The eight fNIRS transmitters were approximately located at AFF4h, F6, AF8, AFp4, AFF3h, F5, AF7, and AFp3 positions, while two receivers were placed at AF7 and AF8 (Schreiner et al. 2021).

account the latency of the hemodynamic response function (HRF) that underlies fNIRS measurements, all binary logistic regression models were limited to 4-6s after stimulus onset, to better capture event related neural activity [49]. Before running the statistical analysis, we calculated the average HbO concentration across all trials in the task and excluded everyone that on average was 2 standard deviations or more away from the group mean as being outliers (which was the case for 4 participants). Separate (identical) models were run for each channel individually to explore the specific contribution of individual PFC regions to memory prediction.

$$\text{mod} = \text{glm}(remembered\_forgotten \sim value,$$
$$\text{data} = data\_file, \text{family} = \text{binomial}) \tag{1}$$

We expected in particular a subsequent memory effect in channels 2 and 6, given that they most likely map onto the inferior frontal gyrus (1).

For the ISC analysis, we followed the methodological approach of [31], calculating inter-subject correlations constrained by subsequent memory. While their analysis was conducted on fMRI data, we adopted it for fNIRS data. Behavioral responses from Question 1, Day 1 were used to create pairwise memory similarity categories. For each stimulus, participant pairs were assigned a value of 1 when both participants correctly remembered the event and 0 when responses differed or participants both forgot. We decided not to include both-forgotten pairs as a standalone category due to an insufficient number of cases. For each participant pair, stimulus, and channel, ISC values were computed by correlating the 9-second HbO time series vectors between the two participants. Correlation coefficients were Fisher z-transformed to improve normality. For each participant pair and channel, z-transformed ISC values were averaged separately across

stimuli belonging to each behavioral category (both remembered vs. different). For each channel, a Bayesian paired-sample t-test was conducted to compare mean ISC values between behavioral conditions using the BayesFactor package in R [50], with a default Cauchy prior (r = 0.707). Bayes Factors (BF) below 0.33 were interpreted as substantial evidence in favor of the null hypothesis [51].

## 3. Results

### 3.1. Behavioral results

Participants' performance on the old/new question of the recognition test was above chance level (M = 0.77, SD = 0.42). Discriminating the old (to-be-recognized) stimuli from the neutral stimuli was near ceiling (M = 0.97, SD = 0.18). Discriminating the old (to-be-recognized) stimuli from the lure stimuli was as expected lower, but still above chance (M = 0.60, SD = 0.49).

### 3.2. fNIRS results

The fNIRS channels were assigned to three regions of interest, where they are likely located: the medial PFC, IFG and dlPFC (1). Due to the shallow depth sensitivity and low spatial resolution of fNIRS, the correspondence between optodes and specific ROIs is approximate and likely only captures partial coverage of these areas (Table 1).

We investigated whether fNIRS measured HbO brain activity during encoding can predict subsequent memory performance. We primarily focused on the channels mapping onto the IFG (i.e., channels 2 and 6), given that a meta-analysis [8] showed this to be the region most consistently mentioned in relation to the subsequent memory effect.

Channel-wise data showed that channel 6 –mapping onto the left inferior frontal gyrus– had significantly higher activity in subsequently remembered compared to subsequently forgotten trials (b = 0.038, SE = 0.016, z = 2.384, P = 0.017, see Fig 3. The right inferior frontal gyrus (i.e., channel 2) did not reach significance (b = 0.006, SE = 0.01, z = 0.635, P = 0.526). Thus, it was in particular the left inferior frontal gyrus that showed a subsequent memory effect in our dataset (i.e., P < 0.025, corrected for multiple comparisons due to separate analyses for left and right hemispheres) (see Table 2).

### 3.3. ISC analysis results

In the ISC analysis we also focused on the IFG. The data analysis showed that that there is no difference in neural synchrony between categories where participants remembered, in comparison to where they remembered differently, or both forgot. Bayesian paired-sample t-tests provided strong evidence for the null hypothesis in channels 2 and 6 (BF10 range = 0.043–0.048), indicating that memory outcome did not modulate neural synchrony regardless of brain region.

**Table 1. Optode, electrode location, hemisphere and likely cortical region. Cortical regions were estimated using the fOLD toolbox v2.2 [52] with the Brodmann atlas for available positions (AF7, AF8, F5, F6). Positions AFp3, AFp4, AFF3h, and AFF4h are not included in the fOLD database and were assigned based on spatial proximity to neighbouring electrodes.**

| Optode | Electrode Location | Hemisphere | Likely Region |
|--------|-------------------|------------|---------------|
| 1 | AFp4 | Right | mPFC / vmPFC |
| 2 | AF8 | Right | Inferior Frontal Gyrus |
| 3 | F6 | Right | Dorsolateral PFC |
| 4 | AFF4h | Right | Dorsolateral PFC |
| 5 | AFp3 | Left | mPFC / vmPFC |
| 6 | AF7 | Left | Inferior Frontal Gyrus |
| 7 | F5 | Left | Dorsolateral PFC |
| 8 | AFF3h | Left | Dorsolateral PFC |

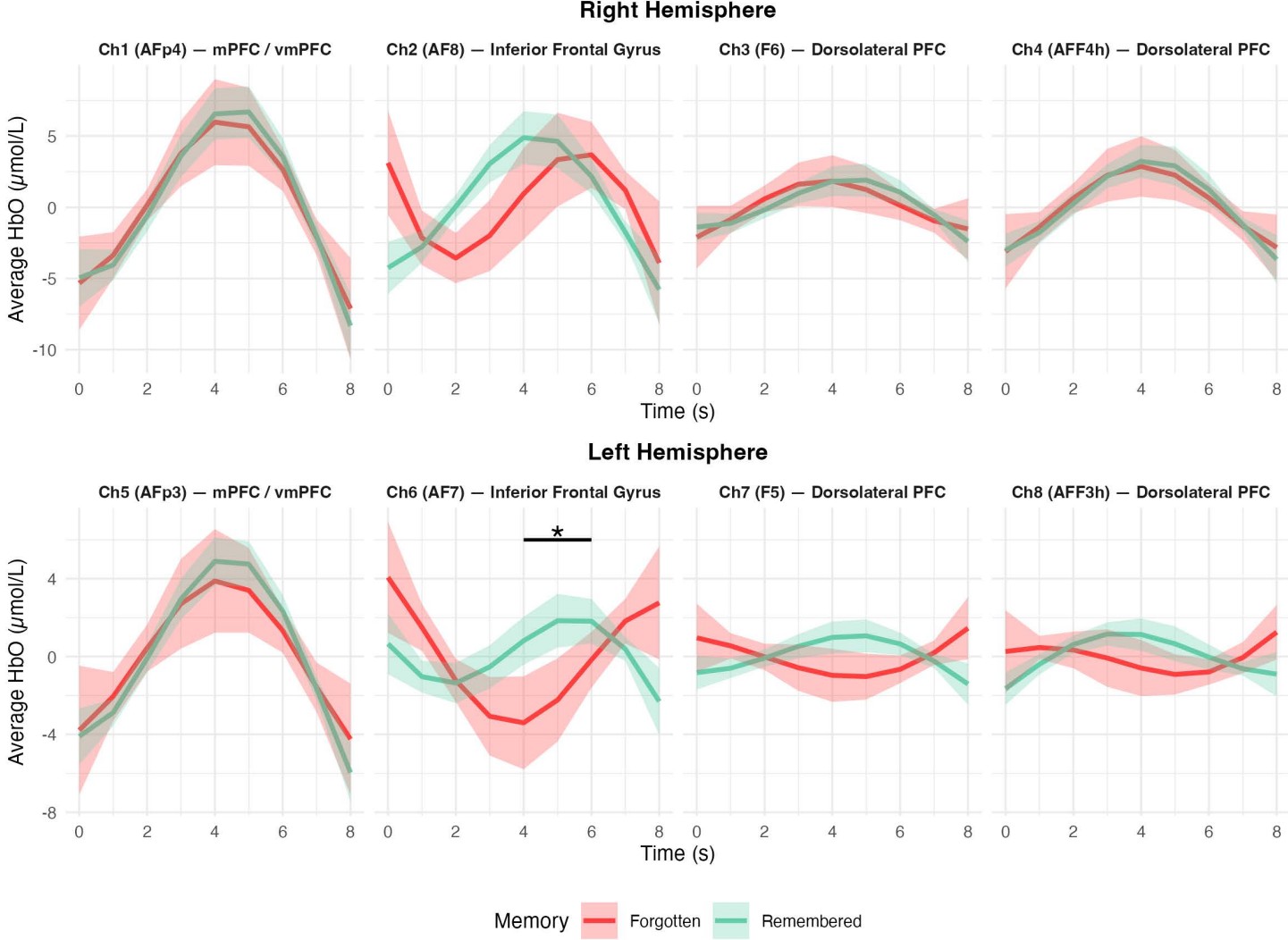

**Fig 3. Channel-wise HbO levels for remembered (green) and forgotten (red) trials (mean±S.E.M.).** Time-point 0(s) is the onset of the stimulus. Channels 1 and 5 map onto the medial PFC, channels 2 and 6 onto the inferior frontal gyrus, and channels 3, 4, 7 and 8 onto the dorsolateral PFC (see [43], Table 1).

## 4. Discussion

Here, we set out to replicate the subsequent memory effect (as previously demonstrated by fMRI) using low-density, portable fNIRS equipment from both an individual and a cross-participant perspective. Our results revealed that it is possible to reliably detect a subsequent memory effect in the left IFG using low-density fNIRS. This suggests that the subsequent memory effect is a robust measurement that is detectable across multiple modalities. On the other hand, the cross-participant analysis did not show evidence for neural synchrony for shared events in the IFG. This suggests that while the IFG reveals differences in neural activity between later remembered and forgotten events, when we look at neural activity between participants, remembered items may be represented by more unique neural patterns in this region. This pattern is consistent with the idea that successful encoding in the IFG may rely on idiosyncratic, individually tailored semantic associations rather than shared neural representations across participants [32]. Altogether, while fNIRS cannot capture

**Table 2. Binary logistic regression results from all channels, filtered between 4-6 seconds.**

| Channel | Estimate | SE | Z | p |
|---|---|---|---|---|
| 1 | 0.0055 | 0.0114 | 0.4857 | 0.6272 |
| 2 | 0.0064 | 0.0101 | 0.6349 | 0.5255 |
| 3 | 0.0093 | 0.0187 | 0.4979 | 0.6186 |
| 4 | 0.0074 | 0.0165 | 0.4480 | 0.6541 |
| 5 | 0.0121 | 0.0146 | 0.8281 | 0.4076 |
| 6 | 0.0378 | 0.0159 | 2.3845 | 0.0171 |
| 7 | 0.0609 | 0.0273 | 2.2299 | 0.0258 |
| 8 | 0.0467 | 0.0275 | 1.6978 | 0.0895 |

the full network of regions across the brain typically observed with fMRI, it may represent a practical and complementary tool for investigating cortical mechanisms of memory in settings where fMRI is not feasible.

The results suggest that IFG, particularly the left IFG, plays a key role in successful memory formation, as indicated by neural differences between later remembered and forgotten events. In particular, at the peak of hemodynamic activation, remembered events elicit higher activity compared to forgotten events. This pattern is consistent with the possibility that the region is involved in the transformation of sensory information into internal representations [3] in the PFC, which may later be consolidated into a longer lasting trace through interactions with the MTL [4]. This interpretation is in line with earlier research suggesting that the IFG is involved in semantic processing [40]. In particular, the IFG has been proposed to contribute during stages of encoding, where it may coordinate semantic representations distributed across the cortex [53]. Furthermore, the IFG has been characterized as a central component of the semantic control system, supporting the controlled retrieval and selection of semantic representations when task-relevant information must be actively accessed [54,55]. Within this framework, the IFG is thought to regulate semantic activation to guide goal-directed processing, forming part of a broader fronto-temporal network supporting semantic comprehension [56]. The present subsequent memory effect in IFG may therefore partly reflect the engagement of controlled or elaborative semantic processing during the comprehension of the event descriptions, such as selecting relevant semantic features or integrating relations among elements of the described situation. Such processes may facilitate the formation of more durable memory traces.

These results are also in line with IFG being considered one of the key brain regions involved in subsequent memory effects [8,10]. The absence of subsequent memory effects in the dorsolateral and ventromedial PFC, regions where fMRI studies have previously reported subsequent memory effects [20,27,57], could partly be explained by the limited coverage and depth sensitivity of low-density fNIRS, making it less suited to detect more distributed or deeper prefrontal contributions. However, this explanation should be interpreted with caution, as these same constraints apply to our IFG channels. It is therefore likely that the IFG finding reflects a genuinely stronger subsequent memory effect in this region, which is in line with meta-analytic evidence identifying it as the most consistently activated region during successful encoding [8]. The trend observed at optode 7 may suggest some dorsolateral PFC involvement, but further investigation with higher-density fNIRS would be needed to draw firm conclusions about this region.

An important contribution of the present study is methodological: our findings demonstrate that the subsequent memory effect can be detected with a low-density fNIRS setup, similarly to another fNIRS study that used the directed forgetting paradigm [39]. While such systems offer limited spatial resolution compared to fMRI, they have clear advantages in terms of cost, portability, and tolerance for movement. The fact that a reliable subsequent memory effect was observed with this setup suggests that fNIRS can be a practical alternative for studying memory processes in contexts where fMRI is not feasible, such as with infants, older adults, or in naturalistic and field settings. Beyond this methodological contribution, the results also strengthen the cross-modal evidence for the left IFG as a robust contributor to successful memory encoding, converging with fMRI and meta-analytic findings [8,10].

## 4.1. Limitations and future directions

Our results only showed evidence for a subsequent memory effect in the left IFG, and not in the right IFG. [8] suggests that there is some evidence that encoding of verbal material is more focused on left hemisphere involvement in the context of subsequent memory analysis, while pictorial stimuli activate more regions in the right hemisphere. Since this experiment included both verbal information (i.e., the event descriptions that the participant read) and pictorial information (i.e., the map with an indication of the specific location), it is difficult to determine whether participants on average focused more on the image or the text. We believe that participants focused more on the text, because they were primarily instructed to remember the story, connect this to a context (holiday or education setting) and only remember the location on the map. They did not have to remember figures, shapes or colors that were presented on the pictures. The fNIRS results also suggest that participants might have focused more on the text. While the behavioral and neural evidence suggests a predominant role of verbal processing, the contribution of spatial encoding cannot be fully ruled out, whether the effect is driven by processing of the spatial component or the verbal descriptions. To make firm conclusions about this lateralization, future research could use an experimental design where verbal and pictorial information are clearly differentiated in time and analyzed separately.

Our results suggest that fNIRS may serve as a viable tool for investigating memory processes. As indicated above, this would be especially relevant in situations where fMRI is impractical, such as with infants, older adults, or in more naturalistic and field environments. However, it is important to note that the present study was conducted with healthy young adults in a controlled laboratory environment, which represents a necessary first step in establishing the feasibility of fNIRS for detecting the subsequent memory effect. Demonstrating reliability under controlled conditions is a prerequisite before translating the paradigm to special populations or more naturalistic settings. For fNIRS to be truly viable outside of laboratory settings, improvements in signal-to-noise ratio and motion artifact handling will also be necessary, as these remain technical challenges in more naturalistic recording conditions. Thus, future research is necessary to translate our subsequent memory effect findings to such naturalistic environments and/or to children and older adults.

## 5. Conclusion

Overall, our findings support the feasibility of using low-density fNIRS to study the subsequent memory effect. Although fNIRS cannot measure the full set of brain regions typically examined when studying the subsequent memory effect with fMRI, it may serve as a practical complementary method for studying cortical mechanisms of memory, particularly in contexts where fMRI is not feasible. Our results also converge with previous findings implicating the left IFG in successful memory encoding, suggesting that this region plays a key role in the early stages of memory formation by coordinating semantic processing during encoding. These findings open possibilities for investigating memory processes in populations and environments where traditional neuroimaging is not practical.

## Acknowledgments

We thank members of the DAF Technology lab at which we ran data acquisition of this project. In particular, we thank Hans van den Dool for help regarding IT and fNIRS hardware/software set-up in the fNIRS lab.

## Author contributions

**Conceptualization:** Silvy HP Collin.

**Formal analysis:** Petra Biro, Silvy HP Collin.

**Project administration:** Silvy HP Collin.

**Software:** Petra Biro.

**Supervision:** Silvy HP Collin.

**Visualization:** Petra Biro.

**Writing – original draft:** Petra Biro.

**Writing – review & editing:** Silvy HP Collin.

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
