## [Decision Letter · Decision Letter 0]

14 Jan 2026

PONE-D-25-53594Subsequent memory effect in the inferior frontal gyrus revealed by fNIRSPLOS One

Dear Dr. Collin,

Thank you for submitting your manuscript to PLOS ONE. After careful consideration, we feel that it has merit but does not fully meet PLOS ONE’s publication criteria as it currently stands. Therefore, we invite you to submit a revised version of the manuscript that addresses the points raised during the review process.

We look forward to receiving your revised manuscript.

Kind regards,

Alberto Dalla Mora, Ph.D.

Academic Editor

PLOS One

Journal Requirements:

Reviewers' comments:

Reviewer's Responses to Questions

**Comments to the Author**

1. Is the manuscript technically sound, and do the data support the conclusions?

Reviewer #1: Partly

Reviewer #2: Yes

2. Has the statistical analysis been performed appropriately and rigorously? 

Reviewer #1: Yes

Reviewer #2: Yes

3. Have the authors made all data underlying the findings in their manuscript fully available?

Reviewer #1: Yes

Reviewer #2: No

4. Is the manuscript presented in an intelligible fashion and written in standard English?

Reviewer #1: Yes

Reviewer #2: Yes

5. Review Comments to the Author

Reviewer #1: In the present manuscript, the authors examine whether functional near-infrared spectroscopy (fNIRS) can capture the well-established subsequent memory effect. By measuring prefrontal activity during encoding, they show that channels over the inferior frontal gyrus (or inferior frontal cortex) exhibit greater activation for items that are later remembered compared to those that are forgotten. The authors claim that their findings suggest that fNIRS can reliably detect subsequent memory–related neural differences, extending this paradigm to populations and settings less accessible to EEG and fMRI.

Nonetheless, despite the clear value of the work, several aspects of the study require further clarification and revision.

Here the relevant points:

Introduction

•Lines 12–17 and 23–33 contain repetition regarding neuroimaging studies investigating the subsequent memory effect.

•Lines 47–50: This section is too vague. It mentions the only fNIRS study cited but does not describe it adequately.

•General questions:

1.Why should it be so difficult to measure the subsequent memory effect, and why is it so important compared to other memory-related tasks?

2.Is the effect tapping into a problem of the “control system” (semantic control)?

•At the end of the introduction, please clarify the theoretical implications: Why is this effect expected to appear in the IFC?

Materials

In general, it is acceptable to refer to a previous work. However, since that work is not yet published, more methodological information should be included in this manuscript (possibly in the Supplementary Materials). Several details are currently missing.

Participants

State at least that participants had no neurological or psychiatric disorders, as this is crucial when measuring memory-related effects.

Stimuli

Are the stimuli balanced for number of words, number of details, etc.? Please provide more information. Ideally, the design should avoid differences in encoding due to low-level content differences.

Task:

•What exactly were participants instructed to do during encoding?

•Is the recognition task sensitive enough?

•How were the filler items for the recognition test created?

These details should appear in this manuscript, not only in the “original manuscript.”

Analysis:

Please describe more clearly your exclusion criteria for participants based on average Hb concentration across trials. Could this issue be addressed through some form of normalization (e.g., mean-centering, z-scoring, baseline correction applied to all trials for each participant), so that participants become more comparable?

Typically, other signal-quality parameters are used when excluding participants—please clarify why this specific choice was made (exlusion based on average HbO). If you don't have a specific reason to do it, I will sugegst to implement one of the normalisation methos stated above, and re-run the analysis with the whole sample.

Results

•At the beginning of the Results you refer to the IFG, while in the Introduction you refer more broadly to the inferior frontal cortex. Please be consistent and justify when you refer to one versus the other.

•Figures: It would be helpful to show the significant results directly in the plots, including the time windows in which significance occurs (e.g., indicated below each subplot). Currently the reader must rely heavily on the text. You might also consider adding labels indicating the anatomical locations associated with each channel, since you refer to them in the text. As they are now, the two figures are overly basic.

•Have you considered implementing decoding analyses to assess whether PFC activity can discriminate between differently encoded stimuli? Time-resolved decoding could strengthen and further support your findings.

Discussion

•The discussion lacks theoretical grounding. How do these results fit within existing memory-related frameworks? Even if the paper is primarily methodological, it should still reference or anchor itself in established theoretical work.

•The limitations regarding null results in dorsolateral and ventromedial PFC are too general and could also apply to your significant results.

•Again, what are the implications of the findings beyond demonstrating that fNIRS can be used in special populations or naturalistic settings? Why is it important to measure the subsequent memory effect specifically with fNIRS?

•The discussion is quite limited, with very few references to relevant literature, both theoretical and methodological.

•You acknowledge the confound of mixing text and images in your stimuli, but this is a substantial issue rather than a minor detail. A deeper discussion is warranted.

•The task has a spatial component because participants had to remember descriptions placed on a map. This is not negligible. Have you considered how spatial encoding might differentially engage the PFC? What is the expected relationship between spatial memory processes and IFG/IFC activity?

•In the limitations, you emphasize that fNIRS could be used in special populations or naturalistic settings, yet your experiment was conducted with healthy adults in a laboratory-like environment. The implications you suggest are therefore not fully supported by the current experimental design and data. Could you elaborate more on this point?

Reviewer #2: This study utilized functional near-infrared spectroscopy (fNIRS) to examine whether the prefrontal cortex exhibits the subsequent memory effect. The results revealed stronger activation in the left inferior frontal gyrus during the encoding of subsequently remembered compared to subsequently forgotten events. This outcome confirms that fNIRS can reliably detect the subsequent memory effect within the left inferior frontal gyrus.

The study is limited by several issues:

The primary limitation is the lack of novel scientific inquiry and theoretical innovation. Although the study employs fNIRS to detect the subsequent memory effect in the inferior frontal cortex, this effect has already been well-established in prefrontal and other brain regions through extensive prior neuroimaging research. Please highlight the novelity in the introduction.

The research question is insufficiently elaborated. Previous studies have identified subsequent memory effects in multiple regions, including bilateral fusiform cortex, hippocampal formation, premotor cortex, and posterior parietal cortex (Kim, 2011). The rationale for focusing exclusively on the prefrontal cortex (PFC), particularly the inferior frontal gyrus (IFG), remains unclear. Moreover, the hypothesis predicts the effect only in the IFG and emphasizes its strength therein, which does not fully address the core question of whether fNIRS can distinguish neural activity during encoding between later remembered and forgotten events.

The experimental paradigm incorporated mixed verbal and pictorial materials, with encoding tasks involving both item memory and associative memory processes. Prior research indicates that subsequent memory effects vary across material types and memory processes (Kim, 2011). The authors should clarify how potential interactions between material type and memory process were controlled to safeguard the validity of the main findings.

The introduction provides an imprecise definition of the subsequent memory effect. Describing it simply as “neural differences at encoding between later remembered and later forgotten events” lacks necessary nuance. As noted in the literature, the term applies specifically when neural activity is greater for later remembered stimuli; the reverse pattern is termed the subsequent forgetting effect (Kim, 2011). A more precise definition should be adopted.

The neural mechanisms underlying the subsequent memory effect are inadequately addressed. The introduction lacks a succinct overview of relevant mechanisms, and the discussion does not interpret the observed activity differences in light of such mechanisms. Furthermore, the study does not meaningfully engage with the opening statement of the introduction regarding why some experiences are remembered and others forgotten. Expanding both the introduction and discussion to incorporate mechanistic perspectives would strengthen the logical coherence of the work.

The justification for citing the fNIRS study by Jing et al. (2022) at the end of the introduction’s second paragraph is unclear and requires elaboration.

No sample size estimation or power analysis was conducted prior to data collection. The authors should demonstrate that the sample size provided adequate statistical power to detect the hypothesized effect.

Several methodological details concerning fNIRS implementation are missing or unclear:

Clarification is needed regarding the term "channel 1–8"—specifically, whether it refers to channels (source–detector pairs) or individual optodes, as misuse of core terminology may cast doubt on the technical rigor of the study (cf. Bíró & Collin, 2025).

Essential hardware and spatial registration parameters must be reported: the number and positions of sources and detectors, channel configurations, and the corresponding brain regions for each channel (preferably with MNI coordinates and Brodmann areas). Their omission undermines the spatial interpretability and reproducibility of the findings.

Key details regarding the analysis time window for hemoglobin concentration data need supplementation:

The unit of time for the selected window ("time 4 to 6") should be specified (e.g., seconds or data points).

It should be explicitly noted that the fNIRS-HbO hemodynamic response function aligns with the fMRI-BOLD response.

Evidence or references supporting the choice of this time window for the memory paradigm used should be provided.

Behavioral results, including the numbers of retained trials for remembered and forgotten conditions, should be reported.

In addition to Channels 2 and 6, results from all channels should be provided in the Results section.

Figures 1 and 2 should include axis labels with appropriate units.

Since channel-wise differences in HbO between remembered and forgotten conditions varied, the rationale for collapsing data across channels for analysis requires justification.

The study identified the subsequent memory effect only in the left IFG, with no effects—either subsequent memory or subsequent forgetting—detected elsewhere. These results do not fully replicate prior fMRI findings. Therefore, the claim that fNIRS can serve as a practical alternative to fMRI in memory research appears premature and insufficiently supported.

6. PLOS authors have the option to publish the peer review history of their article (what does this mean?). If published, this will include your full peer review and any attached files.

Reviewer #1: **Yes:** Giuseppe Rabini

Reviewer #2: No

---

## [Author Response · Author response to Decision Letter 1]

15 Mar 2026

Point-by-point response to the reviewer comments:

Reviewer #1:

In the present manuscript, the authors examine whether functional near-infrared spectroscopy (fNIRS) can capture the well-established subsequent memory effect. By measuring prefrontal activity during encoding, they show that channels over the inferior frontal gyrus (or inferior frontal cortex) exhibit greater activation for items that are later remembered compared to those that are forgotten. The authors claim that their findings suggest that fNIRS can reliably detect subsequent memory–related neural differences, extending this paradigm to populations and settings less accessible to EEG and fMRI.

Nonetheless, despite the clear value of the work, several aspects of the study require further clarification and revision.

Here the relevant points:

Comment 1: Lines 12–17 and 23–33 contain repetition regarding neuroimaging studies investigating the subsequent memory effect.

Thank you for pointing that out. We only mention these studies once now in the introduction.

Comment 2: Lines 47–50: This section is too vague. It mentions the only fNIRS study cited but does not describe it adequately.

We described the mentioned fNIRS study in more detail and why it relates to our study. We also added extra fNIRS studies that investigate memory effects, but not the subsequent memory effect. This part of the introduction now reads as follows:

“fNIRS has been used to measure memory effects, such as episodic memory encoding and recall (Jahani et al., 2017), the effect of age on memory recall in the PFC (Talamonti et al., 2020), how music impacts encoding and recall (Ferreri et al., 2013), influence of working memory related processing (Moya et al., 2025;Geissler et al., 2025; Ren et al., 2025). Jing et al. (2022) measured a construct closely related to SME, using the directed forgetting paradigm, where subjects were instructed to remember or forget a certain stimulus. Behavioral responses were correlated with fNIRS-measured brain activity during encoding trials. No significant differences were observed in the 0–5s window; however, the 5–9s window revealed significantly greater activation for intentional remembering than intentional forgetting in the right angular gyrus. Although this paradigm demonstrates that fNIRS can differentiate between these conditions, to our knowledge, no substantial research has examined the subsequent memory effect using fNIRS.”

Comment 3: Why should it be so difficult to measure the subsequent memory effect, and why is it so important compared to other memory-related tasks?

We extended the theory section to explain this in more detail (see first paragraph of the introduction). Subsequent memory effect is an important concept in memory research, since it can reveal something about a successful memory encoding process through combining brain measures with behavioral evidence. We do not state that the subsequent memory effect is difficult to measure, however, since there is no substantial research on whether fNIRS can show these effects, we believe it is a novel addition to the literature.

Comment 4: Is the effect tapping into a problem of the “control system” (semantic control)?

That is a good point. The IFG is indeed implicated widely in semantic retrieval. For example, regulating access to semantic retrieval especially when retrieval is effortful.

We agree that one possible interpretation of the observed subsequent memory effect might be that successful encoding engaged controlled semantic processing during comprehension of the event descriptions. More specifically, perhaps it reflects selecting relevant semantic features, integrating event elements, or elaborating the meaning of the described situation, which could strengthen memory traces. However, we believe that day 1 of the current paradigm (which data was used) did not explicitly manipulate semantic control demands. Therefore, while IFG activity may reflect controlled or elaborative semantic processing during encoding, the present data cannot specifically attribute the SME to semantic control processes alone. We have clarified this interpretation in the revised discussion.

We added the following paragraph about this interpretation to the discussion:

“This is consistent with earlier research suggesting that the IFG is involved in semantic processing (Poldrack et al., 1999). In particular, the IFG has been proposed to contribute during the early stages of encoding, where it is thought to coordinate semantic representations distributed across the cortex (Gabrieli et al., 1996). Furthermore, the IFG has been characterized as a central component of the semantic control system, supporting the controlled retrieval and selection of semantic representations when task-relevant information must be actively accessed (Badre et al., 2007; Jefferies et al., 2013). Within this framework, the IFG is thought to regulate semantic activation to guide goal-directed processing, forming part of a broader fronto-temporal network supporting semantic comprehension (Lambon Ralph et al., 2017). The present subsequent memory effect in IFG may therefore partly reflect the engagement of controlled or elaborative semantic processing during the comprehension of the event descriptions, such as selecting relevant semantic features or integrating relations among elements of the described situation. Such processes may facilitate the formation of more durable memory traces.”

Comment 5: At the end of the introduction, please clarify the theoretical implications: Why is this effect expected to appear in the IFC?

We included literature on how the initial process of memory formation is expected in the PFC, and more specifically, we included references as to why we expect the effect to appear in the IFG. The end of the introduction now reads as follows:

“We therefore expected the strongest subsequent memory effect in the IFG, given that a large meta-analysis - across as many as 74 neuroimaging studies (Kim et al., 2011) - most consistently implicates this part of the PFC in a subsequent memory effect. IFG is thought to support semantic elaboration during encoding (Poldrack et al., 1999), a process by which new information is linked to existing knowledge and which is considered a key mechanism underlying successful memory formation (Craik et al., 1972). Accordingly, our ISC analysis will also focus on the IFG, investigating whether fNIRS can capture ISC-based SME effects in the PFC, and allowing us to approach the subsequent memory effect from both an individual and a cross-participant perspective.”

Comment 6: In general, it is acceptable to refer to a previous work. However, since that work is not yet published, more methodological information should be included in this manuscript (possibly in the Supplementary Materials). Several details are currently missing.

We provided more information about the study design and stimuli to make it more comprehensible from reading just this article. We hope that these addtiions improved the methods section.

Comment 7: State at least that participants had no neurological or psychiatric disorders, as this is crucial when measuring memory-related effects.

Thank you for pointing that out, we stated this in the subsection about participants.

Comment 8: Are the stimuli balanced for number of words, number of details, etc.? Please provide more information. Ideally, the design should avoid differences in encoding due to low-level content differences.

Before the fNIRS study, we conducted an online pilot study with the same stimuli, without the use of fNIRS, where we only focused on the behavioral results. Here, we conducted a pre-study-survey to validate the stimuli of day 1. The reason for this is that these descriptions should be neutral with regards to their likelihood of being an education activity and being a holiday activity. In this pre-study-survey, participants (N = 6) rated how likely (on a scale from 1 to 5, very unlikely to very likely) people found each of the day 1 activity descriptions to fit with an education setting and separately how likely it would fit with a holiday setting. With this pre-study-survey, we aimed to choose activity descriptions that on day 1 were neutral in their relation towards education and holiday. This would make the association to either education or holiday schema on day 1 purely based on the association to the map that we explicitly indicate for each trial (rather than already inherently in terms of the wording of the descriptions). A score of 3 would be neutral (on a scale from 1 to 5). The used activity descriptions are indeed neutral towards education vs holiday (Figure 1), meaning that a participant that remembers that a given activity description from day 1 relates to holiday is purely based on the in this experiment learned association between that description and the map location as explicitly provided (rather than based on their semantic knowledge about holiday vs education activities).

Figure 1. Original (day 1) descriptions are equally linked to education vs holiday on a scale from 1 (very unlikely) to 5 (very likely).

Further, we computed the number of words in each day 1 activity description and calculated an average number of words for holiday and education descriptions. The average number of words for day 1 indicates a balance in the two categories: 19 for activities in the education categories, and 21 in the holiday category.

To check the difficulty of the stimuli, we computed the mean accuracy for each trial across participants, that remembered a certain specific stimulus for the old/new recognition question on day 1, to determine if any specific stimuli (of the 32 stimuli used) were outliers across the group in terms of how well they in general were remembered (Figure 2). As you can see visualized in the figure, none of the stimuli were significant outliers.

In terms of the visual stimuli (the maps), we used the same map across all stimuli in both the education and holiday context.

Figure 2. Stimulus accuracy on day 1 question 1.

Comment 9: What exactly were participants instructed to do during encoding?

Instruction Resting State Block 1:

To start, you have 3 minutes to relax and prepare for the task. You will receive the instruction for task 1 when that is over.

Introduction:

In the following task you will read descriptions of various activities.

Each activity will be shown along with one out of two maps.

The map indicates at which location that particular activity happened.

Please read the text carefully and try to remember the text and the locations of the activities. Press a key when you are ready to start the experiment.

Instruction Resting State Block 2:

To end part 1 of the study, you have again 3 minutes to relax and prepare for the next task. Please let the experimenter know when this part is over.

Comment 10: Is the recognition task sensitive enough?

We conducted an online pilot study with the same stimuli, without the use of fNIRS, where we only focused on the behavioral results. Although the participants were still above chance for the old/new recognition question of the behavioral pilot study (N = 43), the performance was relatively low. For that reason, we increased the number of repetitions from 3 in the behavioral study to 6 in the fNIRS study. We also decreased the number of descriptions, from a total number of 40 per repetition in the behavioral study to 32 per repetition in the fNIRS study. This indeed led to the desired increase in accuracy, to an average of 77 %.

Comment 11: How were the filler items for the recognition test created?

The filler items (32 lure stimuli and 32 completely new stimuli) were created with chatGPT and manually adjusted. The lure items were manually adjusted to differ by only one word from the original ones, so they differ from the original descriptions in their meaning, which tested if participants paid close attention and were able to memorize the items in detail.

Comment 12: Please describe more clearly your exclusion criteria for participants based on average Hb concentration across trials. Could this issue be addressed through some form of normalization (e.g., mean-centering, z-scoring, baseline correction applied to all trials for each participant), so that participants become more comparable?

Typically, other signal-quality parameters are used when excluding participants—please clarify why this specific choice was made (exclusion based on average HbO). If you don't have a specific reason to do it, I will suggest to implement one of the normalisation methods stated above, and re-run the analysis with the whole sample.

Thanks for bringing this to our attention. Participants whose mean HbO concentration across all trials and conditions was more than two standard deviations below the sample mean were excluded. Such extremely low HbO amplitudes typically reflect poor optode–scalp coupling or low signal-to-noise ratio rather than true neural differences (e.g., due to hair interference or insufficient optical contact). Including these participants can reduce sensitivity because the signal amplitude approaches the noise floor. We do not believe the issue can be circumvented by normalization in this case. Normalization procedures such as mean-centering or z-scoring adjust relative changes within participants but do not improve signal quality when the raw hemodynamic response amplitude is extremely low. In such cases, the signal is likely dominated by noise, and normalization would simply rescale noise rather than recover meaningful neural activity.

Comment 13: At the beginning of the Results you refer to the IFG, while in the Introduction you refer more broadly to the inferior frontal cortex. Please be consistent and justify when you refer to one versus the other.

Thank you for pointing this out. We changed this to consistently state IFG everywhere.

Comment 14: It would be helpful to show the significant results directly in the plots, including the time windows in which significance occurs (e.g., indicated below each subplot). Currently the reader must rely heavily on the text. You might also consider adding labels indicating the anatomical locations associated with each channel, since you refer to them in the text. As they are now, the two figures are overly basic.

We added labels of the corresponding regions, and added significant results on the plot.

Comment 15: Have you considered implementing decoding analyses to assess whether PFC activity can discriminate between differently encoded stimuli? Time-resolved decoding could strengthen and further support your findings.

We thank the reviewer for this interesting suggestion. Decoding analyses could indeed provide additional insights into whether PFC activity discriminates between differently encoded stimuli. However, implementing time-resolved decoding would require substantial additional analyses that fall beyond the scope of the present study. Instead, we performed an intersubject correlation analysis, which allows us to assess the consistency of neural responses across participants during the task and to investigate possible shared neural representations. We have added these results to the manuscript.

Comment 16: The discussion lacks theoretical grounding. How do these results fit within existing memory-related frameworks? Even if the paper is primarily methodological, it should still reference or anchor itself in established theoretical work.

Thank you for this suggestion, we agree that this would strengthen the discussion. We have expanded the discussion to more explicitly anchor our findings in established theoretical frameworks, including Paller's two-component model of memory formation and the prefrontal-MTL interaction framework, and refer back to the introduction where the broader SME literature is discussed. These changes can be found at multiple places embedded throughout the original version of the discussion, which we hope make our findings more theoretically grounded.

Comment 17: The limitations regarding null results in dorsolateral and ventromedial PFC are too general and could also apply to your signific

---

## [Decision Letter · Decision Letter 1]

1 Apr 2026

PONE-D-25-53594R1Subsequent memory effect in the inferior frontal gyrus revealed by fNIRSPLOS One

Dear Dr. Collin,

Thank you for submitting your manuscript to PLOS ONE. After careful consideration, we feel that it has merit but does not fully meet PLOS ONE’s publication criteria as it currently stands. Therefore, we invite you to submit a revised version of the manuscript that addresses the points raised during the review process.

We look forward to receiving your revised manuscript.

Kind regards,

Alberto Dalla Mora, Ph.D.

Academic Editor

PLOS One

Journal Requirements:

Reviewer's Responses to Questions

**Comments to the Author**

1. If the authors have adequately addressed your comments raised in a previous round of review and you feel that this manuscript is now acceptable for publication, you may indicate that here to bypass the “Comments to the Author” section, enter your conflict of interest statement in the “Confidential to Editor” section, and submit your "Accept" recommendation.

Reviewer #1: All comments have been addressed

Reviewer #2: (No Response)

2. Is the manuscript technically sound, and do the data support the conclusions?

Reviewer #1: Yes

Reviewer #2: (No Response)

3. Has the statistical analysis been performed appropriately and rigorously? 

Reviewer #1: Yes

Reviewer #2: (No Response)

4. Have the authors made all data underlying the findings in their manuscript fully available?

Reviewer #1: Yes

Reviewer #2: (No Response)

5. Is the manuscript presented in an intelligible fashion and written in standard English?

Reviewer #1: Yes

Reviewer #2: (No Response)

6. Review Comments to the Author

Reviewer #1: (No Response)

Reviewer #2: In the Discussion and Conclusion section, the authors directly infer that the left IFG participates in semantic processing and contributes to the early stages of memory formation. However, the present study neither directly measured semantic processing nor provided an operational definition of “early stages.” Consequently, this conclusion lacks direct empirical support. While prior literature may aid in interpreting the observed results, it should not be used to draw definitive conclusions. The relevant claims should be revised to prevent overinterpretation.

7. PLOS authors have the option to publish the peer review history of their article (what does this mean?). If published, this will include your full peer review and any attached files.

Reviewer #1: **Yes:** Giuseppe Rabini

Reviewer #2: **Yes:** Mingming Qi

---

## [Author Response · Author response to Decision Letter 2]

14 Apr 2026

Dear Editorial Office of PLOS ONE,

We would like to thank you for the opportunity to submit a revised version of our manuscript. With this revised version, we aim to address the remaining issues as pointed out by the reviewer comments and the journal requirements.

As for the journal requirements listed:

The first round of revisions indeed pointed us to published works (i.e., comment 2-4 of reviewer 2; Kim, 2011). We indeed cite this work as we agree it is very important to this current manuscript. There were no other published works pointed out by reviewers.

We reviewed the reference list and added a few missing volume and page numbers. We believe it is now complete and correct.

As for the remaining reviewer comments:

Comment:

In the Discussion and Conclusion section, the authors directly infer that the left IFG participates in semantic processing and contributes to the early stages of memory formation. However, the present study neither directly measured semantic processing nor provided an operational definition of “early stages.” Consequently, this conclusion lacks direct empirical support. While prior literature may aid in interpreting the observed results, it should not be used to draw definitive conclusions. The relevant claims should be revised to prevent overinterpretation.

We agree with the reviewer that we stated this point too strongly. We adapted this part of the discussion section to the following to soften this claim (see line 245 onward):

“The results suggest that IFG, particularly the left IFG, plays a key role in successful memory formation, as indicated by neural differences between later remembered and forgotten events. In particular, at the peak of hemodynamic activation, remembered events elicit higher activity compared to forgotten events. This pattern is consistent with the possibility that the region is involved in the transformation of sensory information into internal representations (Paller et al., 2002) in the PFC, which may later be consolidated into a longer lasting trace through interactions with the MTL (Simons et al., 2003). This interpretation is in line with earlier research suggesting that the IFG is involved in semantic processing (Poldrack et al., 1999). In particular, the IFG has been proposed to contribute during stages of encoding, where it may coordinate semantic representations distributed across the cortex (Gabrieli et al., 1996).”

We hope that these changes make our manuscript suitable for publication in PLOS ONE.

Sincerely,

Petra Biro and Silvy Collin

---

## [Decision Letter · Decision Letter 2]

28 Apr 2026

Subsequent memory effect in the inferior frontal gyrus revealed by fNIRS

PONE-D-25-53594R2

Dear Dr. Collin,

We’re pleased to inform you that your manuscript has been judged scientifically suitable for publication and will be formally accepted for publication once it meets all outstanding technical requirements.

Kind regards,

Alberto Dalla Mora, Ph.D.

Academic Editor

PLOS One

Additional Editor Comments (optional):

Reviewers' comments:

Reviewer's Responses to Questions

**Comments to the Author**

1. If the authors have adequately addressed your comments raised in a previous round of review and you feel that this manuscript is now acceptable for publication, you may indicate that here to bypass the “Comments to the Author” section, enter your conflict of interest statement in the “Confidential to Editor” section, and submit your "Accept" recommendation.

Reviewer #1: All comments have been addressed

Reviewer #2: All comments have been addressed

2. Is the manuscript technically sound, and do the data support the conclusions?

Reviewer #1: Yes

Reviewer #2: (No Response)

3. Has the statistical analysis been performed appropriately and rigorously? 

Reviewer #1: Yes

Reviewer #2: (No Response)

4. Have the authors made all data underlying the findings in their manuscript fully available?

Reviewer #1: No

Reviewer #2: (No Response)

5. Is the manuscript presented in an intelligible fashion and written in standard English?

Reviewer #1: Yes

Reviewer #2: (No Response)

6. Review Comments to the Author

Reviewer #1: The authors have addressed all the reviewers’ concerns. The manuscript is now more focused and precise regarding both the methodological and theoretical aspects related to the experimental design and research questions.

Reviewer #2: (No Response)

7. PLOS authors have the option to publish the peer review history of their article (what does this mean?). If published, this will include your full peer review and any attached files.

Reviewer #1: **Yes:** Giuseppe Rabini

Reviewer #2: No

---

## [Editor Report · Acceptance letter]

PONE-D-25-53594R2

PLOS One

Dear Dr. Collin,

I'm pleased to inform you that your manuscript has been deemed suitable for publication in PLOS One. Congratulations! Your manuscript is now being handed over to our production team.

Kind regards,

on behalf of

Prof. Alberto Dalla Mora

Academic Editor

PLOS One